# Exploring the Intelligent Emergency Management Mode of Rural Natural Disasters in the Era of Digital Technology

**Jimei Yang [1,*], Hanping Hou [1] and Hanqing Hu [2]**

[1]  School of Economics and Management, Beijing Jiaotong University, Beijing 100044, China; hphou@bjtu.edu.cn
[2]  School of Economics and Management, Beijing Information Science and Technology University,
    Beijing 100192, China; hanqinghu@bistu.edu.cn
*   Correspondence: 18113051@bjtu.edu.cn

**Abstract:** In recent years, rural areas of China have experienced frequent occurrences of various natural disasters. These calamities pose significant threats to the safety, property, and mental well-being of rural residents while also presenting substantial obstacles to the sustainable development of the rural economy. Currently, emergency management in China faces several challenges such as inadequate emergency institutions, insufficient security policies, weak disaster infrastructure, and difficulties in information sharing. In light of this situation, we propose an intelligent command mode based on modern digital technology that capitalizes on its advantages and integrates early warning systems with decision-making processes and rescue operations to establish a comprehensive emergency event processing system. This innovative approach opens up new avenues for exploring and researching effective modes of rural emergency management. The article elaborates on how the construction of a smart rural emergency management mode facilitates the digital integration of disaster elements while enhancing the efficiency of emergency response efforts and promoting sustainable development. The research methodology employed includes literature review methods along with field research techniques and analysis methods. Finally, this discussion evaluates both the benefits and challenges associated with implementing this mode within rural emergency management practices.

**Keywords:** artificial intelligence; big data; emergency management; natural disasters; rural areas

## 1. Introduction

The world geo-climate change and natural variation on the Earth are happening all the time. When these variations bring harm to human society, they constitute natural disasters. Global natural disasters occur frequently, posing great threats to human life and property safety and social economy. For example, in 2004, a 9.3-magnitude strong earthquake occurred in the sea near Sumatra Island, Indonesia, and triggered a tsunami, which affected Indonesia, Thailand, Myanmar, Malaysia, India, and other countries, causing about 290,000 deaths or missing, more than 1 million people homeless, and direct economic losses of USD 4.5 billion [1]. China is one of the countries with the most serious natural disasters in the world. For example, in 2008, the 8.0-magnitude earthquake in Wenchuan, China, caused nearly 69,227 deaths, 17,923 missing, 373,643 injured, and the direct economic loss of CNY 845.215 billion [2]. In the past year of 2023, various natural disasters in China caused a total of 95.44 million people affected, 691 people died or went missing, 3.344 million people were emergency relocated, 2.273 million houses collapsed or were damaged, and 10,539.3 thousand hectares of crops were affected, with direct economic losses of CNY 345.45 billion [3]. It can be seen that the consequences of natural disasters are extremely serious, and natural disasters are also listed as one of the four major public emergencies in China's National Emergency Plan for Public Emergencies. The emergency management work after disasters is facing unprecedented challenges, especially in rural areas, where the imperfect emergency management institution, unintelligent management

method, weak disaster resistance infrastructure, inadequate emergency materials, and unprofessional rescue teams, according to the Global Natural Disaster Assessment Report, 2020, incomplete statistics indicate that rural areas account for over 80% of annual casualties resulting from various natural disasters. Moreover, rural areas also witness more than 90% of fatalities caused by sudden meteorological events like rainstorms, typhoons, landslides, mudslides, and lightning. Additionally, the majority of collapsed houses due to diverse disasters each year are found in rural regions [4]. Consequently, this highlights the urgent need to enhance the construction of intelligent emergency management models in China's rural areas within the context of its rural revitalization driven by artificial intelligence (AI) and big data technologies while simultaneously improving information levels and operational efficiency.

In 2022, the Opinions of the Central Committee and the State Council on Comprehensively Promoting Key Work of Rural Revitalization emphasized the importance of integrating emergency management and rural governance resources in a coordinated manner. It also stressed the need for increased investment in agricultural disaster prevention and reduction. Leveraging digital technologies to enhance rural public services and extend Internet Plus government services to rural areas, thereby advancing the digitization of rural communities. Additionally, we will expand the utilization of big data in agriculture and rural domains to effectively address practical challenges. Furthermore, we will expedite the standardization process for developing digital villages by establishing an evaluation index system for their progress and conducting ongoing trials. Lastly, we will prioritize the development of robust information infrastructure in rural areas. However, China currently lacks an operational unified emergency command and management information system [5]. At present, when dealing with various natural disasters in rural areas, a traditional approach is followed where governments at different levels (prefecture-level city–county (county-level city)–township (town)–village) are involved. This approach adheres to principles such as unified leadership, hierarchical responsibility, comprehensive coordination, classified management, and local administration. Departments report incrementally following a process of reporting–request–approval–issuance–execution. In many regions, there is a lack of synchronization among multi-level government functional institutions which leads to the dispersion of information after disasters occur. This hinders quick sharing and collection of crucial information such as disaster situations, availability of emergency supplies, the status of rescue team deployment, or expert support availability. Additionally, although some emergency management institutions have gradually been established in rural areas, they often operate jointly with government offices where personnel hold multiple positions resulting in insufficient expertise in emergency response knowledge or the adoption of advanced information technology, thereby limiting innovation capabilities. These aspects collectively highlight the deficiencies within the current mode of rural emergency management.

Based on the above problems, this paper explores the optimization of rural intelligent emergency management by integrating five modes under the backdrop of the new generation of information technology. These modes encompass utilizing satellite remote sensing, unmanned aerial vehicle reconnaissance, and ground sensors to achieve comprehensive, multi-level, and high-precision disaster monitoring; establishing a high-speed, stable, and secure communication network for real-time transmission and sharing of disaster information while leveraging big data and cloud computing technologies for intelligent analysis; incorporating machine-learning algorithms to automatically adjust emergency supply scheduling strategies in order to enhance efficiency and ensure accurate and timely delivery to affected areas; comprehensively considering the advantages and limitations of various transportation resources to dynamically deduce hybrid transportation scheduling strategies in complex environments for swift and safe rescue operations; and emphasizing integration and collaboration among diverse rescue resources while optimizing resource allocation through multidimensional fusion to improve effectiveness. The aforementioned five modes converge into a comprehensive, efficient, and intelligent rural intelligent emergency man-

agement system that can provide robust support for rural areas including real-time disaster monitoring with early warning capabilities, rapid response measures along with post-processing work procedures, and effective material allocation coupled with optimized transportation resource utilization, as well as the optimal deployment of rescue teams.

In summary, the research hypothesis of this paper is that we try to prove or deny that the construction of rural intelligent emergency management mode realizes the digital integration of disaster elements, improves the efficiency of emergency management, and promotes economic sustainable development.

A fundamental objective in achieving the rural intelligent emergency management mode is to digitally integrate disaster elements, thereby providing robust data support. Through digital integration, disasters can be comprehended with greater accuracy, early warning and response capabilities can be enhanced, disaster losses can be effectively mitigated, and rural social stability and security can be ensured. Disaster elements encompass various factors that may contribute to disasters, including natural elements such as weather conditions, geology, and the environment, as well as social elements such as population dynamics, the economy, and infrastructure. Digitally integrating these elements entails their conversion into digital formats and employing information technology means for unified management and analysis purposes encompassing data collection, standardization, integration, analysis, and sharing.

The enhancement of the intelligent emergency management mode in rural areas lies in leveraging modern information network technology to achieve the real-time collection, transmission, and processing of disaster information. This enables decision-makers to promptly acquire firsthand information from the disaster scene and make swift decisions, thereby reducing the decision-making time and enhancing efficiency. Moreover, the intelligent emergency management mode also employs big data analysis and artificial intelligence technology to facilitate intelligent scheduling and the optimal allocation of rescue resources. Various aspects including rescue personnel, materials, and equipment are comprehensively considered to ensure their prompt arrival at the disaster site. Simultaneously, through establishing a unified information platform and implementing a data sharing mechanism, different departments can collaborate across boundaries while staying updated on each other's progress and resource requirements. Such coordination forms a formidable force that enhances the overall emergency response efficiency.

The rural intelligent emergency management mode not only enables an effective response to natural disasters but also facilitates the sustainable development of rural areas through long-term monitoring and planning. For instance, by analyzing influential factors such as climate change and natural disasters, it becomes possible to formulate rational agricultural production plans and implement ecological protection measures that enhance the quality of the ecological environment and economic development level in rural regions.

This paper establishes a solid theoretical foundation for emergency management, expanding its theoretical framework while enhancing the scientific nature of emergency decision-making. The construction of the rural intelligent emergency management mode contributes significantly by providing a comprehensive and scientifically grounded system specifically tailored for rural areas. This mode takes into account the unique characteristics and complexities associated with these regions, integrating information technology with emergency management practices to achieve intelligent, precise, and efficient outcomes. Moreover, it enhances the scientific basis of emergency decision-making processes. Traditional modes of emergency management often rely on experience and subjective judgment, making them susceptible to decision errors or delayed responses. In contrast, the rural intelligent emergency management mode employs modern information technology and data mining techniques to conduct more accurate analysis and prediction regarding potential emergencies. Consequently, this approach improves both accuracy and timeliness in making critical decisions during emergencies while effectively reducing disaster-related losses.

## 2. Literature Review

Scholars both domestically and internationally have conducted extensive research in the relevant fields mentioned above. Regarding the new generation of information technology, Shannon (1948) posited that the value of information lies in reducing uncertainty and increasing certainty, aligning perfectly with the essence of emergency management [6]. Information technology serves as a crucial technical guarantee for achieving the modernization and intelligence transformation of emergency management. Chung et al. (2022) proposed that leveraging modern information technology is beneficial for planning and implementing emergency management efforts. Emerging technologies such as the Internet, geographic information systems (GIS), remote sensing, satellite communications, radar communications, mobile communications, and wireless network systems can provide support for establishing an infrastructure for emergency management information [7]. Yang et al. (2017) [8] and Bartoli et al. (2015) [9] suggested that smart cities' platforms and adopted modern technologies can assist in all stages of emergency management work by enabling prediction, monitoring, and response to emergencies. Adam et al. (2020) believed that the new generation of digital technology will revolutionize traditional decision-making modes in emergency management. Sensor networks are pivotal technologies within the Internet of Things paradigm offering novel research directions for urban emergency management [10]. Li et al. (2012) mentioned that the Internet of Things connects people, objects, and the Internet more accurately [11], and its key technologies include RFID technology, sensor network technology, intelligent technology, etc. Franca et al. (2021) found that the Internet of Things plays a positive role in disaster relief, rescue, response, and recovery and can promote communication between professionals, provide reliable data and more accurate prediction, personnel tracking, material and drug distribution, etc., [12]. Zhang et al. (2013) believed that the computer accessed the server of the rescue Internet of Things gateway through the HTTP protocol using the IE browser and displayed the rescue request information sent by the front-line rescuers of the earthquake rescue in real time on the command center computer. At the same time, the command center could send response signals to the rescue handheld terminals of the rescue personnel at the scene who sent the request through the IE browser [13]. Guo et al. (2022) believed that the Internet of Things technology could monitor natural disasters in real time in the emergency management platform and develop early warning plans and rescue measures scientifically and reasonably according to the information collected. The Internet of Things technology could also be used to improve the intelligence of rescue equipment and docking the real-time operation data of rescue equipment with the background database of emergency management to realize the remote intelligent command and scheduling of rescue equipment [14]. Cloud computing has super data processing speed, which can process tens of thousands of data in a few seconds. Cloud computing can store and calculate massive emergency data in the emergency management platform and can realize real-time monitoring. It can also realize remote collaborative office through the cloud platform, and all personnel are not restricted by geographical location and can communicate and share information in real time, greatly improving the efficiency of emergency management. Big data are a huge collection of data, which far exceeds the processing capacity of traditional database software in acquisition and analysis. At the technical level, big data and cloud computing are often inseparable, and big data must be processed by distributed architecture relying on cloud computing. Jahangiri et al. (2017) further studied the positive role of big data in reducing human suffering and economic losses caused by natural disasters [15]. And Daniel et al. (2013) used big data technology to quickly grab and intelligently analyze massive information to provide decision support for natural disasters [16]. Zhou et al. (2021) elaborated that in the complex processing process of big data, artificial intelligence, cloud computing, deep learning, machine learning, and other digital technologies are needed to support [17]. Gong et al. (2022) stated that in the emergency response and disposal stage, artificial intelligence technology can realize automatic disaster response reporting through geographic information data and social media network data fused with deep-learning technology and realize an emergency

evacuation route map through satellite image fusion image-recognition technology and machine-learning technology [18]. Therefore, algorithms and artificial intelligence technologies will be employed in the development of rural emergency management models. Zhu et al. (2021) utilized blockchain technology for monitoring and early warning in an intelligent emergency system. Upon activation of the early warning contract, real-time sharing of original data and epidemic synchronous detection can be achieved through network broadcasting, transforming the traditional hierarchical early warning arrangement into a more flexible geared approach. This enables organizations at all levels to promptly access early warning information simultaneously and respond instantaneously [19]. Furthermore, Fosso's research (2022) also demonstrates that information and communication technology can enhance public awareness and mitigate the adverse impacts of natural disasters [20].

In terms of emergency management, Zhou (2023) asserts that significant attention has been given to frontier technologies such as 5G, artificial intelligence, big data, and blockchain within government planning and policy frameworks [21]. Zhang (2022) analyzes the strategic role of new-generation information technology in emergency management [22]. Shah (2019) illustrates how integrating the Internet of Things with big data facilitates disaster emergency response systems for smart cities [23]. Najafi et al. (2020) proposed establishing a comprehensive earthquake emergency response framework that proactively responds by uniting rescue forces from various sources under unified command to achieve interconnection and the sharing of disaster-related information [24]. Al-Kaff et al.'s proposal (2020) involves employing autonomous unmanned aerial vehicles (UAVs) for forest fire emergency rescue operations [25]. Sabrina (2016) conducted a study on the challenges faced by local governments at the grassroots level in emergency management, emphasizing the significance of focusing on the pivotal role played by social media in emergency response and recovery. Additionally, an evaluation framework was developed [26]. Hamad (2023) proposed a novel framework that utilizes classification algorithms to process perception data, relies on the Internet of Things, and integrates multiple machine-learning models for the real-time collection of severe weather data that may lead to floods. This framework also sends early alerts to users and rescue teams in case of any potential threats [27]. Gething et al. (2011) suggested a theoretical approach for constructing disaster emergency management capacity by combining data collection and analysis with real-time decision-making processes. They further recommended using mobile phone network data to track population flow during emergencies, enabling prompt rescue operations and timely treatment for affected individuals [28]. George et al. (2017) examined necessary changes required for future emergency management implementation in the United States. They identified challenges and opportunities across five areas: climate change, leadership, funding, communications and emerging technologies, and partnerships, as well as cultural promotion [29]. Seaberg et al. (2017) analyzed game theory extensively within the context of disaster response stages but proposed its extension into other phases such as disaster reduction, preparedness, and recovery [30]. Tennakoon et al. (2021) investigated an emergency management framework specific to a region in New Zealand and discovered that small-scale recurrent disasters were overlooked when formulating policies related to emergency management. Therefore, they advocated incorporating these impacts into policy systems to eliminate disparities in risk management at lower administrative levels [31]. In terms of the emergency response in rural areas, Korkmaz (2023) discovered that post-earthquake outbreaks of lice and scabies infections often arise due to overcrowding and inadequate sanitation in rural regions. Tetanus and its associated complications also frequently occur as a result of insufficient or lacking vaccination [32]. Additionally, Chan (2019) identified distinct healthcare needs for rural areas following natural disasters compared to urban areas [33]. Liu (2013) and Ning (2013) argued that numerous rural communities in certain Asian countries have a significant proportion of elderly individuals and children, necessitating increased attention towards specific health risks during emergency management. These risks include respiratory problems caused by indoor waste pollution resulting from energy poverty [34,35]. Hurst (2024) demonstrated that the provision of financial resources

during natural disasters can enhance the mental health and well-being of rural victims [36]. Regarding legislation and regulations, Funta (2023) examined the necessity for regulation within the digital context [37], Skora (2022) discussed administrative justice reform's role in Poland during the COVID-19 pandemic [38], while Popa Tache (2023) proposed the theoretical possibility of establishing an interdisciplinary legal mechanism through pooling key capabilities [39].

Through the aforementioned literature analysis, it is evident that current research predominantly focuses on the integration of new-generation information technology in government planning and the strategic analysis of emergency management, primarily within the context of smart cities. However, there is a noticeable lack of emphasis on rural informatization. Furthermore, model construction primarily revolves around the singular perspective of emergency material provision, with limited attention given to the application of digital technologies such as artificial intelligence, big data, and cloud computing throughout the entire process of rural emergency management. Therefore, this paper presents an innovative comprehensive emergency management model encompassing early warning, command scheduling, material allocation, vehicle coordination, and rescue team deployment.

## 3. Aim, Materials, and Methods

This paper aims to comprehensively investigate how the rural intelligent emergency management model, constructed within the context of digital technology, facilitates the digital integration of disaster elements, enhances emergency management efficiency, and fosters sustainable development.

Scientific literature encompasses textual, numerical, visual, auditory, and other forms of data that document research achievements and academic perspectives. It possesses significant reference value, practical utility, and scholarly merit. Apart from peer-reviewed journal articles, academic monographs, conference papers, and dissertations, it also includes informal publications like technical reports, experimental records, and scientific research data. Scientific literature serves as a means for researchers to comprehend the latest trends in their field of study while acquiring research methodologies and exchanging viewpoints. By perusing scientific literature, researchers can grasp advancements in related domains thereby providing theoretical foundations and technical support for their work. Theoretical data entail information derived from existing theories or conceptual frameworks which establish a theoretical groundwork for comprehending subject matter while furnishing an analytical framework for interpretation. It is grounded on established principles and concepts within the pertinent academic or professional sphere. Theoretical data refers to information obtained from prevailing academic or professional fields with the purpose of constructing analytical frameworks and theoretical underpinnings. These data typically rely on well-established principles and concepts encompassing noteworthy research accomplishments, as well as ideas within the discipline's purview. They may exist in various formats sourced from diverse channels such as academic institutions, government departments or enterprises, etc. To acquire and organize such data necessitates employing multiple methods including literature retrieval techniques along with knowledge mapping approaches, etc. Theoretical data play a crucial role in research, as they facilitate comprehension of fundamental concepts and pivotal issues within a given subject, while also providing an analytical framework for problem-solving.

The literature research method involves the systematic collection, organization, analysis, and interpretation of relevant literature materials to achieve specific research objectives. This approach is widely employed across various disciplines and relies on logical reasoning and dialectical analysis to explore the research subject from diverse perspectives, multiple levels, and various dimensions in order to unveil its internal relationships, essential characteristics, and developmental patterns. In this study, we focus on emerging areas such as the new generation of information technology, rural emergency management, and intelligent perception, among others while consulting both domestic and interna-

tional literature sources. By thoroughly reviewing the existing theoretical foundations and research achievements in these fields and analyzing their current status quo along with future development directions, we identify our own research questions based on previous studies while highlighting deficiencies within existing theories and practices. Simultaneously adopting a holistic and systematic perspective, this paper proposes the establishment of a rural intelligent emergency management model that elucidates how the digital integration of disaster elements can enhance emergency management efficiency, thereby promoting sustainable development. The field investigation method entails researchers personally visiting the site to engage in activities such as observation, interviews, and other investigations. This approach allows for the collection of first-hand data and information, which is crucial in uncovering the essence and patterns of the problem at hand. In this study, the author conducted a comprehensive survey within rural seismic belt areas by collaborating with local emergency management departments, transportation authorities, hospitals, seismological experts, and meteorological experts. The aim was to gain insights into the challenges faced by rural areas during natural disaster response processes, particularly regarding issues related to the implementation of new-generation information technology. Based on these practical problems identified through analysis, further research was conducted. Analytical methodology is a systematic and organized research approach aimed at conducting in-depth analysis and comprehension of diverse information sources. By breaking down complex elements into more manageable components, this method facilitates understanding and examination. It involves evaluating the relationships between these components and drawing logical conclusions based on a comprehensive understanding of each part's significance and interconnections, thereby providing robust support for problem-solving endeavors. Synthetic methodology represents an integrated process that necessitates the harmonious integration of various elements or ideas from multiple perspectives to generate a unified and nuanced outcome. Rather than simply summarizing individual parts, synthesis entails uncovering internal connections and interactions through profound comprehension and analysis. Its objective is to enhance the depth of knowledge by amalgamating different aspects of the subject matter to yield novel insights or perspectives. Comparison serves as an essential method focusing on identifying similarities and differences among research elements or cases. This approach enables us to comprehend problems from multiple perspectives and levels, thus offering additional ideas and solutions for effective problem-solving strategies [40].

## 4. Intelligent Emergency Management Mode of Rural Natural Disasters under the Background of Digital Technology

Through literature analysis and field research, significant disparities in intelligent emergency management modes between rural and urban areas in China have been identified, primarily manifested in the emergency management approach, infrastructure, information technology, emergency resources reserves, information dissemination, and public awareness of disaster response, as illustrated in Table 1.

**Table 1.** Comparison table of urban and rural emergency management modes.

| Urban Emergency Management | Rural Emergency Management |
|---|---|
| modernization of management approach<br>advanced infrastructure<br>modern information technology<br>rich emergency resources reserves | backward management approach<br>poor infrastructure<br>low level of information<br>less emergency resource reserves |
| diversified ways of information dissemination | single mode of information dissemination |
| strong public awareness of disaster response | poor public awareness of disaster response |

China's focus on the development of rural emergency management information is evident in eight national documents. These include the Opinions of the CPC Central Committee and the State Council on Comprehensively Promoting Rural Revitalization in 2022 (Zhongfa (2022) No. 1), the National 14th Five-Year Plan and Long-term Objectives Outline for 2035, as well as plans such as the National Emergency System Plan for the 14th Five-Year Plan, Development Plan for Radio, Television, and Internet Audiovisual Services in the 14th Five-Year Plan, and the Development Plan for National Emergency Broadcast System Construction in the 14th Five-Year Plan. The objective is to promote coordinated integration of rural emergency management resources and governance, accelerate the construction of active release terminals for rural emergency broadcasting, and provide guidance on safe evacuation during emergencies. Emergency broadcasting plays a crucial role in disseminating national policies, managing emergencies effectively, governing society, and constructing spiritual civilization by bridging information gaps at local levels while ensuring precise mobilization. It also facilitates effective communication channels that foster innovative development within rural spiritual civilization construction efforts. In line with this emphasis on rural revitalization, Gansu Province has issued its own opinions advocating accelerated progress in constructing rural emergency management information systems based on implementing Spirit of Central Document No. 1 in 2022 according to recommendations from the CPC Emergency Management Committee. China aims to empower rural emergency management through digital technologies by extending smart fire control measures into rural areas while also expediting the implementation and utilization of an information-based monitoring system.

According to the theory of emergency management, this paper establishes the framework of a rural intelligent emergency management model. The principles of emergency management and collaborative governance theory are seamlessly integrated, effectively harnessing the strengths of government, medical institutions, enterprises, and other departments to achieve synergistic effects. Drawing on the key concepts from crisis theory regarding reduction force, we construct the fundamental framework for a rural natural disaster intelligent emergency management model. This includes an integrated intelligent perception mode encompassing both aerial and ground-based observations, a command and scheduling mode that leverages communication networks enhanced by multiple elements, an emergency material scheduling mode based on machine learning algorithms, and a dynamic deduction approach for hybrid transportation scheduling under complex disasters, as well as a rescue team scheduling mode that integrates multidimensional resources (Figure 1).

The fundamental architecture of the intelligent emergency management mode for rural natural disasters encompasses the integrated intelligent perception mode of "sky and ground", the command and scheduling mode that enhances communication network plurality, the machine learning-based emergency material scheduling mode, the dynamic deduction-based hybrid transportation scheduling mode under complex disasters, and the multi-dimensional resource integration-driven rescue team scheduling mode, as depicted in Figure 1.

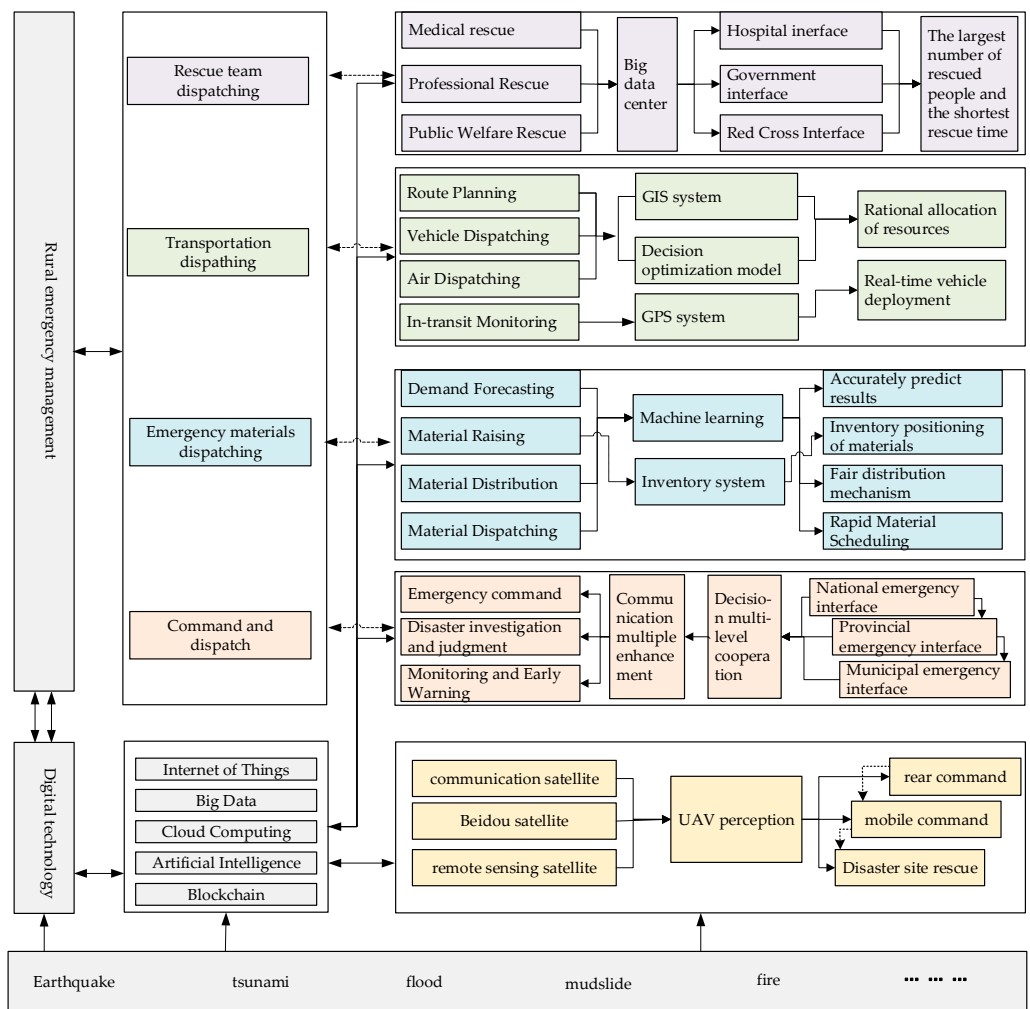

**Figure 1.** Basic architecture of intelligent emergency management mode for rural natural disasters. Source: Own processing according to the literature.

*4.1. Integrated Intelligent Perception Mode of "Sky Air and Ground"*

Integrated sensing mode is an advanced technology independently developed by China for the monitoring and warning of natural disasters. This technology combines positioning and communication technologies, such as communication satellites, Beidou satellite communication, and remote-sensing satellites, to achieve the network interconnection of satellite communication, UAV sensing, and ground command systems. By applying this technology, the disaster site can be comprehensively, deeply, and real-time sensed and monitored. Additionally, the "Sky–ground" integrated sensing mode integrates multiple services including satellite portable stations, remote-sensing monitoring, and Beidou navigation communication to enhance the efficiency and accuracy of disaster monitoring. Through this mode, real-time information about the disaster site can be obtained including topography data analysis along with rescue progress updates enabling the visual monitoring of on-site conditions. This mode not only provides robust support for rural natural disaster analysis but also offers theoretical insights into important technical challenges like intelligent perception (Figure 2).

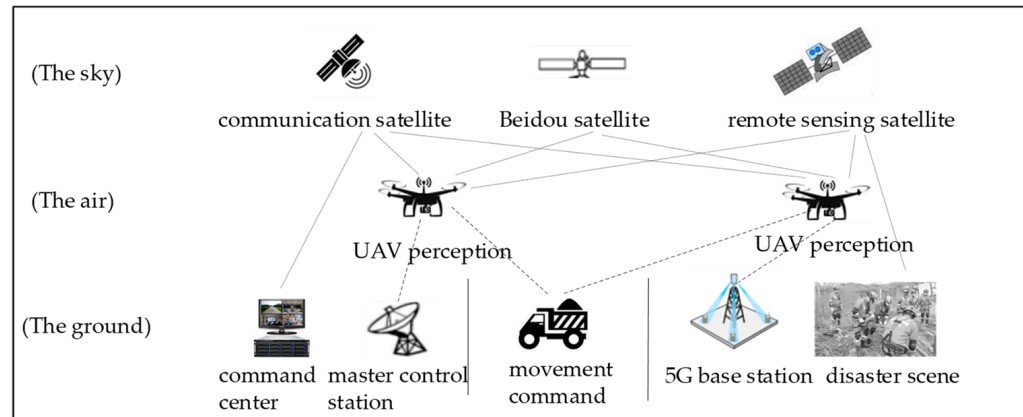

**Figure 2.** Integrated intelligent perception mode of "sky air and ground". Source: Own processing according to the literature.

### 4.1.1. Sky Sat Satellite Communication System

The Sky Sat satellite communication system comprises a diverse range of satellites, encompassing various domains such as communication, Beidou navigation, remote sensing, and Earth observation. In the event of natural disasters like earthquakes, floods, and landslides, the Sky Sat satellite communication system can promptly initiate an emergency response mechanism by deploying relevant satellites for the real-time monitoring of the affected areas. By assimilating data from multiple sources, it enables accurate assessment of the disaster's scale, impact extent, and potential casualties. Moreover, this system can also relay crucial disaster information to unmanned aerial vehicles (UAVs) and other sensing devices through network transmission. This approach significantly enhances decision-makers' comprehension regarding the comprehensive nature and precision of on-ground conditions in disaster-stricken regions while providing a scientific basis for effective rescue operations and post-disaster reconstruction efforts. To summarize its applications thus far: the Sky Sat satellite communication system has been employed in fields including natural disaster monitoring, environmental surveillance, as well as public safety measures. Consequently, rural emergency management should draw insights from these cases and experiences to expedite the development of natural disaster information systems.

### 4.1.2. "Airborne" UAV Perception System

The "Airborne" man-machine perception system is an intelligent technical system that integrates advanced sensors and high-speed data processing technology. In the context of disaster relief, this system receives a vast amount of information from the "Sky" satellite communication system, including images, sounds, temperature, humidity, and pressure. It then transmits real-time situational updates regarding building damage status and road conditions, as well as the number and locations of trapped individuals in the disaster area. This invaluable information provides crucial support to rescue personnel by offering timely, accurate, and comprehensive intelligence assistance to the ground command system.

### 4.1.3. "Ground" Command System

The "Ground" command system not only receives real-time data from the airborne UAV system but also transmits these crucial data to the rear command platform, front mobile command system, and on-site rescue personnel. This enables decision-makers to quickly obtain an accurate understanding of the situation and develop more scientific and efficient rescue strategies. During actual rescue operations, the UAV provides aerial surveillance of the entire disaster area and relays high-definition images in real time. Equipped with various sensors, it collects environmental data such as temperature, humidity, and gas concentration. The "Ground" command system integrates and analyzes this information to facilitate prompt decision-making. Specifically, in earthquake or mudslide scenarios and

other disasters, the "Ground" command system assists in assessing damage severity and establishing rescue priorities by prioritizing the areas most severely affected for immediate assistance. Simultaneously, the rear command platform coordinates resources including materials, equipment, and manpower to provide robust support for on-site rescue efforts. The front mobile command system closely collaborates with on-site rescue teams while adjusting plans in real time to ensure an orderly and efficient execution process.

### 4.2. Multi-Enhancement Command and Scheduling Model for Communication Network

The multi-dimensional enhancement of command and dispatch mode encompasses crucial components such as monitoring and early warning, disaster research and assessment, emergency command, and subsequent post-disaster reconstruction and experience analysis. Firstly, in terms of monitoring and early warning, advanced communication technologies including the Internet of Things, big data, and artificial intelligence are utilized to real-time collect diverse disaster information, providing accurate and timely risk notifications for government departments, as well as the general public. Secondly, regarding disaster research and assessment, various types of disasters along with their impact scope and severity are swiftly identified by integrating all forms of monitoring data alongside on-site information. This is accomplished through the comprehensive utilization of technical means such as communication networks, geographic information systems (GIS), expert systems, etc., thereby offering a scientific foundation for subsequent emergency command.

Subsequently, in terms of emergency command procedures, the command and dispatch system is employed to transmit on-site information promptly while ensuring seamless communication between emergency management departments at all levels including rescue teams as well as affected individuals. Additionally, rational rescue strategies along with resource allocation plans are formulated based on distinct characteristics exhibited by different disasters coupled with on-site conditions so as to enhance the overall rescue efficiency, as depicted in Figure 3.

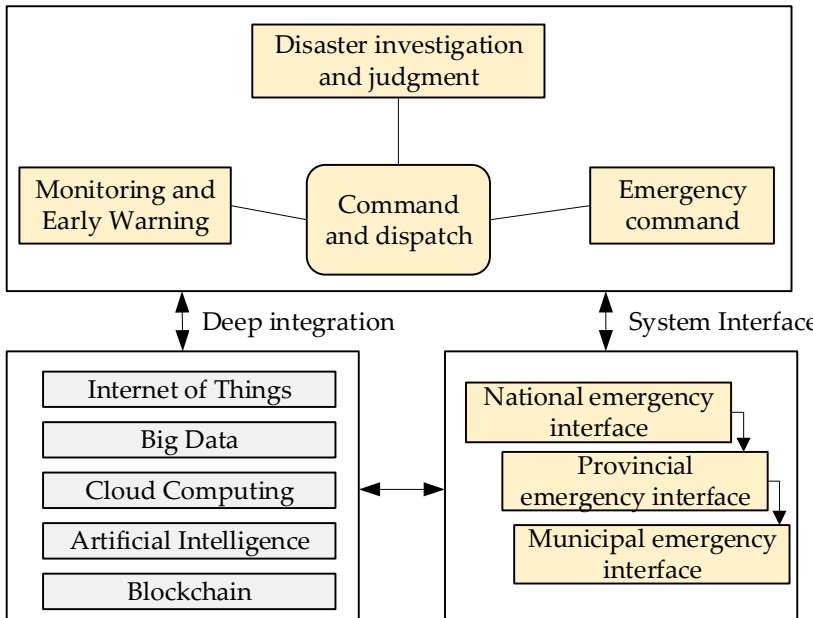

**Figure 3.** Multi-enhancement command and scheduling model for communication network. Source: Own processing according to the literature.

#### 4.2.1. Early Warning and Monitoring

In the early warning and monitoring stage, advanced technologies such as satellite remote sensing, unmanned aerial vehicle remote sensing, global positioning system (GPS), Beidou positioning system, geographic information system (GIS), and Internet of Things

sensors are employed to accurately measure and comprehensively observe various regions of the Earth's surface from multiple perspectives. Satellite remote-sensing and unmanned aerial vehicle remote-sensing technologies offer a wide range of geospatial information facilitating the rapid assessment of overall area conditions; the global positioning system (GPS) and Beidou positioning system ensure the precise positioning of monitoring targets for data accuracy; the geographic information system (GIS) integrates and analyzes these vast datasets to provide valuable reference information. Furthermore, Internet of Things sensors enable the real-time monitoring of environmental changes in specific areas such as temperature, humidity, and pressure for more accurate prediction of the weather fluctuations and natural disaster risks. Following the collection of this extensive information, the command system employs data processing and analysis techniques to extract valuable insights that serve as a scientific foundation for subsequent early warnings and decision-making.

### 4.2.2. Disaster Research and Decision-Making

Through the utilization of received early warning information, a combination of big data processing and expert support facilitates scientific disaster research and decision-making, thereby providing a foundation for emergency command. The collected early warning information undergoes rapid analysis, mining, and processing using advanced big data techniques to ascertain the correlation, impact magnitude, and occurrence probability among different types of disasters. This analysis identifies key areas and time periods that may pose threats to human lives and property while offering pertinent decision-making support for subsequent disaster research and judgment processes. Furthermore, relevant departments coordinate with experts in meteorology, geology, and water conservancy to consult on and evaluate the outcomes derived from big data analysis. Leveraging their extensive experience and professional knowledge enables more accurate forecasting of disaster development trends while delivering comprehensive research findings to emergency commanders.

### 4.2.3. Emergency Command

Emergency command utilizes cutting-edge information technologies including integrated remote sensing, 5G technology, big data analytics, GIS (geographic information system), artificial intelligence (AI), and digital twin simulation to facilitate disaster monitoring, analysis, decision-making, planning generation, and command execution. This is achieved by integrating unmanned aerial vehicles (UAVs), remote-sensing analysis techniques, video surveillance [41], and other technical equipment into the rural natural disaster monitoring system; leveraging GIS geographic information technology and AI recognition algorithms for the comprehensive collection of disaster site resources and data; employing big data analytics for intelligent analysis in conjunction with duty systems and expert support systems to holistically assess the disaster situation; utilizing a natural language processing-based AI model [41,42] to formulate emergency command plans; harnessing 5G technology for real-time information sharing with the platform enabling command vehicles, as well as individual digital devices and other sensing devices on-site to establish an interactive real-time digital twin scene [43]; and thereby assisting in command decision-making processes while achieving accurate and timely emergency command execution.

### 4.3. Scheduling Model of Emergency Supplies Based on Machine Learning

The scheduling model of emergency supplies based on machine learning includes the demand forecasting of emergency supplies, raising of emergency supplies, allocation of emergency supplies, and scheduling of emergency supplies, as shown in Figure 4.

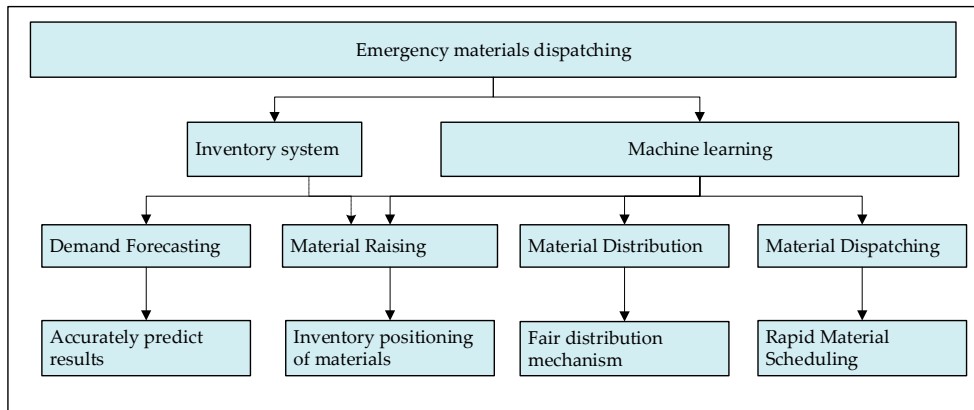

**Figure 4.** Scheduling model of emergency supplies based on machine learning. Source: Own processing according to the literature.

### 4.3.1. Emergency Supplies Demand Forecast

In the aftermath of a disaster, the availability of emergency materials is crucial for ensuring basic survival and sustaining the lives of victims. Therefore, it is imperative to establish an efficient emergency material management system in rural areas with limited information resources and inadequate material reserves. This system should encompass four key modules: forecasting emergency material demand, collecting emergency materials, distributing them effectively, and scheduling their delivery.

The forecasted amount of required emergency materials refers to the quantity determined by decision makers using machine-learning techniques and various forecasting models to meet the basic life necessities of victims post-disaster. Accurate forecasting serves as a foundation for the effective collection, scheduling, and distribution of these essential supplies. Overestimating forecasts can lead to redundant allocation of scheduling resources and wastage of emergency materials; conversely, underestimating forecasts may result in insufficient distribution within disaster-stricken areas, thereby jeopardizing victim's lives. Leveraging advancements in new-generation information technology will provide more efficient support for accurate forecasting by utilizing big data mining techniques, artificial intelligence algorithms, satellite remote sensing technologies, etc., enabling real-time prediction and monitoring capabilities regarding material requirements during disasters [43,44].

### 4.3.2. Emergency Supplies Raising

The process of emergency supplies inventory management involves accurately locating the material inventory by integrating the emergency management platform of government departments at all levels, based on the predicted quantity of emergency supplies required after a disaster occurs. These reserve materials are primarily stored by government entities, enterprises, and power generation facilities. Additionally, there are social donation materials that utilize RFID technology to store information and update it after each stage. Blockchain technology is employed to ensure the security of material information and data.

### 4.3.3. Emergency Supplies Distribution

The distribution of emergency supplies entails promptly matching the appropriate quantity and category of emergency materials to each affected area within a limited timeframe following a disaster occurrence. In dealing with complex and extensive data related to emergency supplies, big data technology provides an environment and foundation for their distribution, enabling decision-making processes to shift from being "experience-driven" to "data-driven" [45,46].

### 4.3.4. Emergency Supplies Scheduling

The scheduling module for emergency supplies integrates cloud computing technology, effectively incorporating numerous distributed emergency resources into the platform. It intelligently assesses the demand for emergency supplies during unforeseen natural disasters while precisely adapting available resources according to predictions made by the system regarding raised and distributed emergency supplies. This facilitates efficient resource allocation across regions, platforms, and departments while enhancing overall rescue efficiency [47].

### *4.4. Mixed Transport Scheduling Model under Complex Disasters*

The transportation management system primarily encompasses functions such as transport route planning, vehicle scheduling, and on-the-way monitoring. Leveraging technologies like RFID and GPS from the Internet of Things (IoT), available vehicle resources are collected [48], while real-time information regarding vehicles, materials, and disaster areas is integrated with the transport planning system through network technology. By employing cloud computing, data mining, and other advanced techniques, intelligent analysis is conducted on the aforementioned vehicle resources, human resources, and material resources to devise optimal emergency material transport routes. Through communication technology and GIS technology integration, real-time position updates for transportation vehicles along with transport status information and road conditions are transmitted back to the transportation management system in order to achieve the real-time monitoring of transportation vehicles, as well as tracking management of emergency materials, while providing feedback on optimal paths among other relevant details (Figure 5).

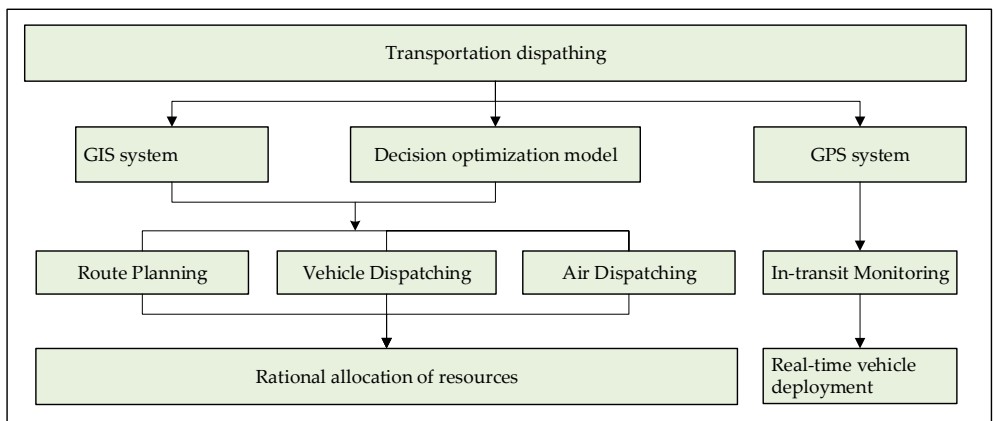

**Figure 5.** Mixed transport scheduling model under complex disasters. Source: Own processing according to the literature.

### 4.4.1. Transportation Route Planning

In the face of sudden disasters or emergencies, emergency management departments encounter significant challenges in efficiently and expeditiously transporting emergency materials to affected areas. By leveraging Internet of Things technologies such as radio frequency identification (RFID) and the global positioning system (GPS), real-time deployment vehicle resources in and around the affected areas can be collected and integrated into a central database. Through network technology, real-time information including vehicle resources, material demand, and the status of affected areas can seamlessly connect with the transportation planning system to ensure the optimal selection of material transport routes. Furthermore, advanced technologies like cloud computing and data mining can be employed for comprehensive intelligent analysis on acquired vehicle resources, human resources, and material resources. For instance, historical data analysis enables the prediction of the required types and quantities of materials in the upcoming hours while facilitating reasonable resource allocation accordingly. Simultaneously, by analyzing

vehicle driving trajectories along with speed data, scheduling strategies can be optimized to ensure efficient and secure material transport processes. Based on these considerations combined with the transportation planning system's capabilities, an optimal route for emergency material transport can be formulated that accounts for real-time traffic conditions as well as road conditions while addressing supply urgency and disaster-stricken area needs—thereby ensuring the prompt delivery of emergency supplies to support effective disaster relief efforts.

### 4.4.2. Vehicle Scheduling

The integration of modern communication technology, geographic information system (GIS) technology, and traffic interconnection technology enables real-time transmission of vehicle position, transport status, road conditions, and other Upon careful reading data. This provides accurate and comprehensive support to the transportation management system while enhancing the rationality and efficiency of vehicle scheduling. Communication technology facilitates real-time communication between on-site personnel and the transportation management system during emergency rescue operations, enabling prompt feedback on vehicle position and status for scheduling commands. GIS technology processes and analyzes geographic information to provide relevant data such as road conditions and traffic flow to assist in planning and optimizing transportation routes for improved efficiency. Furthermore, through integrating various types of transportation vehicles' information within a global scope, the entire transportation network can be effectively scheduled and controlled. Additionally, interconnection technology can be employed in emergency rescue scenarios to promote collaborative operation among vehicles for enhanced efficiency and accuracy.

### 4.4.3. On-the-Way Monitoring

Following a disaster, China promptly deploys rescue vehicles to the affected area with utmost urgency. These vehicles are equipped with state-of-the-art rescue equipment and experienced personnel, poised to provide immediate emergency assistance to those affected. To ensure the safe and efficient operation of these vehicles, a real-time monitoring system is installed onboard. By continuously monitoring transmitted data from the equipment, timely updates on vehicle status, speed, position, and other relevant information can be obtained for effective guidance and dispatching purposes. Furthermore, this monitoring system also enables the real-time tracking of crucial parameters such as fuel levels and tire pressure to guarantee that emergency situations do not compromise the effectiveness of rescue efforts.

### 4.5. Rescue Team Scheduling Mode of Multi-Dimensional Resource Integration

The management of rescue teams primarily encompasses various types of personnel involved in post-disaster rescue operations, including experts, professional rescuers, medical staff, and volunteers engaged in public welfare rescue efforts. By establishing a connection with the expert database within the emergency management system at all levels, experts can be accurately matched to provide professional support to local government emergency management personnel. Professional rescue teams consist of emergency response personnel maintained by government departments such as firefighters. In cases where local professional rescuers are insufficient, cross-zone support can be requested through the local emergency management platform, and the intelligent allocation of rescue personnel can be facilitated through system platform analysis. Medical staff are intelligently allocated to designated disaster areas based on actual conditions such as the extent of the disaster, number of casualties, and severity of injuries or illnesses. Public welfare rescue personnel engage in voluntary activities and their information is stored within a dedicated database that reserves a certain number for daily availability. These individuals undergo certification and training conducted by government departments [48]. During times of disasters, they are allocated accordingly based on specific rescue requirements (Figure 6).

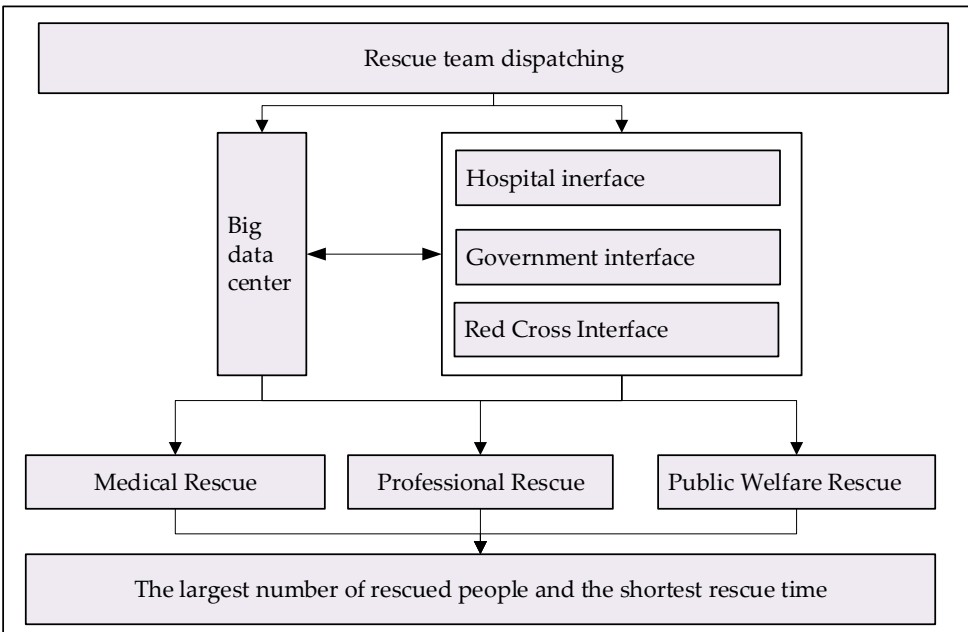

**Figure 6.** Rescue team scheduling mode of multi-dimensional resource integration. Source: Own processing according to the literature.

### 4.5.1. Medical Rescue Team

In the event of a natural disaster, relevant agencies in China will immediately start the emergency plan according to the actual situation, casualties, and injuries. The system will intelligently assign a certain number of medical personnel to the affected area for rescue. These medical personnel usually come from cross-regional medical institutions across the country. They will gather and form a rescue team in the initial time after the disaster, carrying necessary medical equipment and medicine to the affected area. In the rescue process, medical personnel not only need to provide on-site first aid to the injured, but also need to classify the condition and transfer the seriously injured to nearby hospitals for further treatment. In order to ensure the smooth development of rescue work, China will also mobilize various resources to provide necessary living materials and infrastructure support to the affected area in the initial time after the disaster. For example, the rapid establishment of temporary hospitals, provision of clean drinking water and food, and guarantee of the normal operation of power and communication facilities will be paid attention to and implemented.

### 4.5.2. Professional Rescue Team

In China, frequent occurrences of various natural disasters such as earthquakes, floods, and landslides pose significant threats to the safety of people's lives and property. Professional rescue teams possess extensive experience in emergency response. In situations where local professional rescue forces are insufficient due to disasters in a specific area, the government can promptly request cross-regional support through the local emergency management platform. Upon receiving requests from the cross-regional emergency platform, relevant departments intelligently allocate rescue personnel based on analysis results obtained from the cross-regional platform. These highly skilled and experienced personnel are swiftly deployed for rescue operations. During cross-regional support efforts, an intelligent distribution system optimizes resource allocation to enhance overall efficiency; for instance, by analyzing team capabilities and disaster severity across different regions, resources are allocated reasonably to prevent wastage while ensuring timely and effective support is provided to each affected region.

### 4.5.3. Public Rescue Team

Public rescue personnel refers to organized volunteers actively participating in emergency rescue operations. These personnel form a nationwide volunteer network, and the talent information database platform has emerged as a crucial tool for better organizing and coordinating them. This platform centralizes the management of personal information, expertise, and experience of public rescue personnel across the country. However, it is not only important to have an adequate number of teams but also essential to ensure their professional quality and emergency response capabilities. Therefore, government departments are responsible for certifying and training these volunteers. During disasters, suitable public education teams will be allocated from the platform based on specific situations and demands.

### 5. Discussion

The intelligent emergency management model based on digital technology has facilitated the digital integration of disaster elements, thereby enhancing the efficiency of emergency management and fostering sustainable development in rural economies within China's rural areas. In the subsequent discussion and conclusions, we will delve into the advantages brought forth by this model, as well as the challenges and opportunities encountered during its implementation.

Based on the scientific literature, government documents, field research, and analysis, this paper proposes a rural intelligent emergency management model guided by the principle of "intelligent perception and intelligent emergency". Firstly, to enhance the information level in rural areas, it is crucial to leverage new-generation information technology. By establishing a comprehensive intelligent perception system (i.e., integrating aerial and ground monitoring), round-the-clock monitoring and perception of rural areas can be achieved, effectively improving their information collection and processing capabilities. Simultaneously, developing a diversified communication network command and scheduling system enables the rapid and accurate transmission of information as well as efficient command and coordination during various emergencies. Secondly, to improve the efficiency and accuracy of emergency response efforts, an emergency supplies scheduling system based on machine learning can predict the required supplies for different types of disasters using real-time information for effective scheduling. Additionally, a dynamic deductive transportation scheduling system can perform optimal path planning and scheduling based on real-time road conditions and material needs to significantly enhance material transportation efficiency while ensuring timely delivery. Finally, in enhancing rescue capacity, a multi-dimensional resource fusion rescue team scheduling system can be employed to coordinate and optimize the allocation of various rescue forces.

The objective of this model is to establish a diversified technology chain for addressing emergency rescue by integrating multi-dimensional data on rural natural disasters and the latest generation of information technology. In order to enhance the efficiency of dealing with rural natural disasters, this model employs satellite remote-sensing technology and big data analysis methods to assess the disaster situation, ensuring the real-time comprehension of dynamic disaster events. Simultaneously, it optimizes command and scheduling through artificial intelligence technology in order to achieve the comprehensive monitoring and cross-regional coordination of emergency personnel, equipment, and material resources within the affected area. Furthermore, it utilizes machine-learning algorithms for material allocation in response to real-time changes in the disaster situation, thereby improving resource optimization for rescue teams. This approach not only enables swift responses to rural natural disasters but also enhances rescue efficiency while minimizing the losses incurred. Consequently, this model will contribute towards enhancing rural areas' capacity for preventing natural disasters, safeguarding citizens' lives and property safety as well as providing robust support for sustainable development efforts. Additionally, it will facilitate widespread adoption of cutting-edge information technologies while promoting economic society transformation and upgrading.

The promotion of rural intelligent emergency management mode is confronted with a series of formidable challenges. Firstly, there exists a substantial disparity between rural infrastructure construction and the requirements of this mode, which proves arduous to bridge within a limited timeframe. This discrepancy encompasses not only hardware facilities such as communication networks and intelligent devices but also software conditions like technical personnel and data resources. The absence of these prerequisites will directly impede the implementation and operation of the rural intelligent emergency management system. Secondly, the intelligent emergency management mode involves multidisciplinary technological innovation research in areas such as artificial intelligence, big data, the Internet of Things, etc., which are unevenly distributed across rural regions in China. On one hand, advanced technology may lack applicable scenarios in these areas; on the other hand, it may fail to meet the actual needs of rural communities. This imbalance in technological development poses difficulties for promoting the rural intelligent emergency management mode effectively. Thirdly, there is an absence of technical and managerial standards in its application that can lead to confusion during the implementation process of the rural intelligent emergency management system and hinder achieving the desired outcomes. Lastly, government regulation also presents challenges to adopting and supporting this mode due to various reasons resulting in uncertainty that can impact both its speed of promotion and successful implementation.

In the future, the construction and implementation of rural intelligent emergency management mode need to be addressed from multiple perspectives. Firstly, it is crucial to prioritize disaster infrastructure and information construction, including establishing a robust disaster warning system and enhancing monitoring and forecasting capabilities in the meteorology, geology, and hydrology fields. Additionally, strengthening the information infrastructure development in rural areas is essential. Secondly, promoting the comprehensive integration of theory with cutting-edge technologies such as artificial intelligence, big data, and the Internet of Things (IoT) should be emphasized for improving efficiency and accuracy in emergency management. Simultaneously, scientific research must be reinforced to establish a scientific and standardized management system tailored to rural characteristics. Furthermore, during the construction process of a rural intelligent emergency management mode, attention should also be given to verification procedures along with standardization efforts supported by appropriate policies. Prior to implementation, thorough verification tests and pilot studies are necessary to ensure feasibility while emphasizing the development of standardized work processes and operating procedures for enhanced efficiency. Moreover, governments should increase investment in this mode while encouraging social resources' participation through pluralistic investment mechanisms.

## 6. Conclusions

The information level in rural areas of China is generally inadequate, particularly concerning the management of natural disasters, which becomes notably prominent. Due to the limited availability of information, rural areas often encounter a range of practical challenges during natural disasters, including outdated emergency management methods, deficient command systems, and low rescue efficiency. To address these issues, this paper explores an emergency management model for rural natural disasters and investigates how emerging technologies such as the IoT, big data analytics, cloud computing, AI, and blockchain can be integrated with emergency management to establish a suitable mode for addressing rural natural disasters. Within this framework, rescue personnel can be rapidly and effectively deployed; timely identification and delivery of emergency supplies to affected regions can be ensured; and transportation vehicles will securely transport essential resources. Consequently, the digital integration of disaster elements is achieved while enhancing the efficiency of emergency management practices. This approach promotes sustainable development within rural economies by extending the theoretical foundation of emergency management and improving the scientific nature underlying decision-making processes. In general, the academic community has conducted extensive research on in-

telligent emergency management in the context of digital technology; however, it still encounters numerous challenges. When constructing an intelligent emergency mode, it is crucial to fully consider the unique characteristics of rural areas, such as their complex geographical environment, inadequate infrastructure, and difficulties in information dissemination. Additionally, there is a need to enhance information technology training for rural residents and improve their information literacy so that they can effectively enhance mutual assistance and self-help capabilities during natural disasters. The limitation of this paper lies primarily in its theoretical nature. Future research should focus on integrating various natural disaster characteristics while deeply incorporating new-generation information technology with emergency management theory and practice to further enrich and transform theoretical research outcomes. In summary, constructing a rural natural disaster emergency management mode is a comprehensive project that requires leveraging emerging technologies' advantages while continuously exploring and innovating based on the actual situation in rural areas to provide robust support for promoting digitalization and modernization towards preventing and mitigating natural disasters.

**Author Contributions:** J.Y. designed and wrote this paper; H.H. (Hanqing Hu) participated in the investigation; H.H. (Hanping Hou) provided valuable research insights into the analysis and structure of the paper. All authors have read and agreed to the published version of the manuscript.

**Funding:** This research was funded by the Ministry of Education of Humanities and Social Sciences project under Grant No. 21YJA630029.

**Data Availability Statement:** Data are contained within the article.

**Acknowledgments:** The authors would like to thank the anonymous peer reviewers and editors for their comments.

**Conflicts of Interest:** The authors declare no conflicts of interest.

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
