# Peer review of "Exploring the Intelligent Emergency Management Mode of Rural Natural Disasters in the Era of Digital Technology"

_sustainability, doi:10.3390/su16062366_

Round 1

Reviewer 1 Report

Comments and Suggestions for Authors

This is an interesting paper and sound research.  The paper should provide a clear methodology section on the literature review approach. The diagrams are good but it should be clear if these are sourced or adapted from literature or whether they are the work of the authors.

Also, it would strengthen the paper to make a clearer distinction between the existing practices in urban areas and the constraints in rural areas, in line with the title and rationale of the paper. 

Comments on the Quality of English Language

The quality of the paper is good. The arguments are sound and the topic very interesting.  With a little more clarity in writing, the paper could be improved and make a good contribution to literature.

Author Response

Dear reviewer,

The article has undergone significant revisions in the following aspects:

1. The key words have been alphabetically sorted.
2. The introduction now includes the research hypothesis.
3. The literature review incorporates an analysis of frontier literature in the related field and highlights the innovation of this article compared to existing studies.
4. Additional chapters on research methods and research objectives have been included.
5. Chapter 4 provides a comprehensive comparison between urban and rural emergency management, supplements theoretical foundations, demonstrates the feasibility of our research model through government documents, and includes proper citation for all graphs used.
6. The discussion section effectively addresses the research hypothesis.
7. In conclusion, we reflect upon limitations encountered during this study using past tense language.
8. Typos and abbreviations of proper nouns within the paper have been corrected.

Please refer to the attached document for a detailed version of these revisions.

Thank you very much for your valuable feedback during review process, and we eagerly await your response.

Reviewer 2 Report

Comments and Suggestions for Authors

The article is of a theoretical and cognitive nature. The issue is important and current on a global scale. The issues discussed in the article in relation to the current state of knowledge are innovative.
The presented solutions are based on new technologies taking into account various aspects. The indicated solutions based on new information technologies require empirical verification, which in turn requires further research. The authors write about this important aspect in the discussion.
The discussion of the results is actually a summary of the proposed solutions. This part of the article should be considered as requiring correction. The general conclusions from the conducted research should be compared with the achievements of the literature. The literature review also requires correction. Although the bibliography contains 33 items, the list lacks many current studies related to the topic of the article.
The article draws attention to the development imbalance between urban and rural areas. This is a very important aspect in the context of currently promoted development concepts based on the principles of cohesion.
The 'intelligent crisis management mode in rural areas' was presented in terms of available tools and proposed solutions. The practical dimension of the article should be highly appreciated. However, the typically scientific side has been somewhat neglected. There is a lack of research hypotheses. They need to be formulated. They should then be referred to in the discussion of the results. You should also indicate how the article complements the current state of knowledge.
The above comments should be taken into account in the new version of the article.

Author Response

(The authors gave the same response as above.)

Reviewer 3 Report

Comments and Suggestions for Authors

The manuscript is well written, but there is nothing numeric and there are not any specific data on hazard and natural disasters. Everything is descriptive. The authors have not performed any mathematical models and coding. I would recommend carrying out research with more concrete methodology, data results and conclusion. I don’t any other comments except the one given below.

1   Line 61: What is CPC?

Author Response

Dear reviewer,

The article has been modified in the following aspects:
1. I apologize for the incorrect CPC and have made necessary revisions.
2. This paper incorporates a wider range of literature research methods, field research methods, and analysis techniques. It validates the feasibility of the proposed model through examination of actual government documents and enhances the mathematical model in subsequent sections.
3. The keywords have been alphabetically sorted.
4. A research hypothesis has been added to the introduction section.
5. The literature review now includes an overview of frontier literature in the related field as well as highlighting the innovations introduced by this paper compared to existing studies.
6. New chapters on research methods and research objectives have been included.
7. Chapter 4 provides a comparative analysis of existing urban and rural emergency management systems, supplements theoretical foundations, highlights contributions, verifies the applicability of our proposed model using government documents, and adds proper citations for all graphs used.
8. The discussion section addresses each research hypothesis individually.
9. In conclusion part, we have written in past tense while elaborating on limitations encountered during this study's execution.
10.Typos and abbreviations for proper nouns throughout the paper have been corrected.

Please refer to attachment for detailed revised version.Thank you very much for your valuable feedback during review process; we eagerly await your response.

Reviewer 4 Report

Comments and Suggestions for Authors

The scientific study as a whole is really very interesting in terms of content. However, authors must also adhere to the systematic and content structure required by the scientific journal itself published on the website.

It is not enough to state in the abstract that the article contains 31 improvements, the most significant contributions must be included in the study to appeal to the readers.

Sort keywords alphabetically.

In the introduction, the research questions and/or hypotheses must be stated clearly and unambiguously, to which the authors must also answer in the final sections.

The chapter "2. Literature Review" must be strengthened especially with other relevant literature from the Web of Science and SCOPUS databases. Only 33 sources are really not enough for a scientific article in this type of journal.

Attention should also be focused on the development of smart cities and the importance of digital connectivity in the process of building smart cities. A topic could be the WiFi4EU initiative, which provides funding for free public WiFi networks and supports digital inclusion. The basic pillars of smart cities are also important, including digital connectivity, efficient transport, environmental protection, innovation and citizen participation. Cooperation between city authorities and local communities is vital to achieve successful digital development of not only cities but also the countryside, which will improve protection against unwanted situations.

Current scientific works such as e.g.

1. Kaššaj M, Peráček T. 2024. Sustainable Connectivity—Integration of Mobile Roaming, WiFi4EU and Smart City Concept in the European Union. Sustainability, 16 (2):788. https://doi.org/10.3390/su16020788

2. Funta, R.; Buttler, D. The Digital Economy and Legal Challenges. InterEULawEast 2023, 10 (1), 145–160, doi: 10.22598/iele.2023.10.1.8

3. Skora, A; Srebalova, M and Papacova, I. 2022. Administrative judiciary is looking for a balance in a crisis. Juridical Tribune, 12 (1), pp. 5-20, doi : 10.24818/TBJ/2022/12/1.01

Popa Tache, C.E., & Săraru, C.-S. (2023). Lawfare, Between its (Un)Limits and Transdisciplinarity. Precedente Revista Jurídica, 23, 37-66. https://doi.org/10.18046/prec.v23.5889

The description of scientific research methods should be in a separate chapter. This part is "scattered" throughout the manuscript. This part needs to be completely finished. Attention should also be paid to the description of the individual scientific research methods used, such as analysis, description, synthesis, comparison. The aforementioned authors Kaššaj & Peráček 2024 appropriately address this issue.

Last but not least, the set research questions/hypotheses must also be clearly answered in the discussion/conclusion.

The last chapter 5. Conclusion must be written in the past tense and should contain findings, not a description of what the individual chapters were about. Briefly address the limitations of current research.

Author Response

(The authors gave the same response as above.)

Round 2

Reviewer 3 Report

Comments and Suggestions for Authors

The manuscript is good.

Author Response

Dear reviewer,

I sincerely appreciate your valuable feedback in Round 1.

Reviewer 4 Report

Comments and Suggestions for Authors

Dear authors,

I thoroughly read your extensively edited manuscript. I expected a better description of the changes made. You forgot to express why you did not accept the literature suggested by me.

Among other changes, you added source 33, which I did not propose to add. This source (Benčo, J. 2021) can be found in the References of the recommended work Kaššaj & Peráček 2024 under no. 23. I can't imagine how you could quote from it, since it is not available in electronic form on the web, and in order to be able to quote it, you have to have access to it, it is not enough just to take it from another work. And this procedure makes me worry about the ethical side of your work. In addition, you copied the surname of the author of the work incorrectly.

It would also be advisable to focus on a more scientific style of writing the manuscript, just passively referring to sources is not enough. I recommend proceeding as follows: According to Skora et al., 2022..., ... We agree with the opinion of XY ...2021 ..

Again (even if I don't like it) I state that there is a disproportionate preponderance of works by Chinese authors in the work, the balance of the literature used is lacking. The number of sources (40) is still insufficient for the scope of such a scientific study. Resources need to be replenished.

A small reminder at the end. You have completed the abstract, but it is extremely long, it should contain a maximum of 200 words, so that it is clear, concise and especially concise.

I am also impatiently waiting for the changes to be made.

Author Response

Dear reviewer,

I sincerely apologize for the negligence in properly citing Kashšaj & Peraček 2024 under no. 23. Going forward, I will ensure to be more diligent in addressing such issues. I have thoroughly reviewed the literature you recommended, which has significantly contributed to the development of this paper and has been appropriately referenced in both the literature review and methods section. Based on your feedback, this round of revisions primarily focuses on refining the references and abstract section. All modifications have been highlighted with yellow underlines for your convenience.Please refer to the attached document for detailed information, and the modifications are outlined below.

1. The abstract section is succinct and concise, providing a condensed overview of the study.
2. The reference format has been revised to adhere to XY et al.'s (2022) findings...
3. Irrelevant literature has been excluded, including 13 Chinese sources, while 23 English publications in this field have been incorporated.

Round 3

Reviewer 4 Report

Comments and Suggestions for Authors

Dear authors,

I am glad that you accepted my comments, which resulted in raising the scientific level of your manuscript to the required level.

Based on the changes made, I agree to publish the manuscript.

Reviewer